∂ | **Open Peer Review** | Bacteriology | Research Article

# Bacterial skin colonization with a specific *Cutibacterium avidum* clade as a risk factor for periprosthetic joint infections—a multicenter study

Llanos Salar Vidal,[1,2] Julia Prinz,[3] Pascal M. Frey,[4,5] Tiziano A. Schweizer,[3] Laura Böni,[4] Silvio D. Brugger,[4] Holger Brüggemann,[6] Jaime Esteban,[1,2] Yvonne Achermann,[3,4,7] on behalf of the ESGIAI (ESCMID Study Group for Implant-Associated Infections)

**ABSTRACT** *Cutibacterium avidum* is increasingly recognized as a causative agent of periprosthetic joint infections (PJIs), yet data on its pathogenic potential and distinguishing features from commensal strains remain limited. In this multicenter study, we compared 11 *C. avidum* isolates from PJIs with 32 isolates from healthy skin collected across four European hospitals. We investigated phylogenetic relationships, antibiotic susceptibility, biofilm formation, and bacterial fitness. Phylogenomic analysis revealed two main clades within the *C. avidum* population. All PJI isolates belonged exclusively to Clade 1, which also included skin isolates. Within Clade 1, gene content analysis showed no consistent genetic differences between PJI and skin isolates. All isolates exhibited moderate to strong biofilm formation, with no significant differences in either data set. Minimal inhibitory concentration (MIC) and minimal biofilm inhibitory concentration (MBIC) values were low and largely concordant, while minimal biofilm eradication concentration (MBEC) values were elevated for all antibiotics except rifampin. One isolate was resistant to clindamycin due to the *erm(X)* gene. Rifampin consistently showed the lowest MIC, minimal bactericidal concentration, MBIC, and MBEC values. Bacterial fitness, assessed via bacterial quantitative fitness analysis, was significantly lower in PJI isolates compared to skin isolates when all strains were analyzed ($P = 0.039$), but this difference was not statistically significant when restricted to Clade 1. In conclusion, *C. avidum* isolates are strong biofilm producers irrespective of clinical origin. PJI isolates are restricted to a single phylogenetic clade, yet lack distinct biofilm or fitness traits within that clade, suggesting that multiple Clade 1 strains may have the potential to cause PJIs.

**IMPORTANCE** *Cutibacterium avidum* has long been considered a skin commensal, but it is increasingly associated with prosthetic joint infections (PJIs). Despite its clinical emergence, little is known about its virulence potential or how invasive strains differ from commensal ones. This multicenter study provides the most comprehensive comparative analysis to date, integrating phenotypic and genomic data from both PJI-associated and skin-derived isolates. We show that all isolates are strong biofilm formers and that invasive isolates exhibit reduced growth fitness—a phenotype linked to persistence and treatment failure in other pathogens. Notably, all PJI isolates belonged to a single phylogenetic clade, suggesting that specific lineages of *C. avidum* may be more likely to cause infection. These findings help clarify the biology of this emerging pathogen and provide a foundation for improved diagnostics, susceptibility testing, and future infection prevention and treatment strategies.

**KEYWORDS** *Cutibacterium*, prosthetic joint infection, biofilm, antimicrobial susceptibility

Address correspondence to Yvonne Achermann, Yvonne.Achermann@usz.ch.

Llanos Salar Vidal and Julia Prinz contributed equally to this article. The author order was determined by consensus among all authors.

Holger Brüggemann, Jaime Esteban, and Yvonne Achermann contributed equally to this article.

The authors declare no conflict of interest.

See the funding table on p. 11.

Infections related to arthroplasty cause high morbidity and also impose an increased cost to the healthcare system. The most common bacteria isolated from prosthetic joint infections (PJIs) are staphylococci, followed by streptococci, enterococci, gram-negative bacilli, and anaerobes (1, 2).

Even though anaerobic bacteria make up only approximately 3–6% of PJI (3), they can cause more than 50% of infections in shoulder implants (4). Among them, more than 70% are gram-positive bacteria belonging to *Cutibacterium* spp. (5) with *Cutibacterium acnes* monopolizing the attention (6). However, another species from this genus, *Cutibacterium avidum*, is also gaining importance (6–8).

*C. avidum* is a member of the human skin microbiota. It tends to reside in wet areas such as the axilla, nares, groin, and rectum (7). It is considered a skin commensal with low virulence potential; however, it can act as an opportunistic pathogen in superficial and deep/invasive infections such as skin abscesses, abdominal infections, breast infections, infective endocarditis, prostate infections, and bone and joint infections (8–11). *C. avidum* is an infrequent etiological agent of PJI, predominantly in late chronic infections. It preferentially colonizes moist areas; hence, mainly obese individuals who underwent primary hip arthroplasty were found to be affected (8, 12, 13).

Many different virulence factors may be involved in PJI, including biofilm formation (7). However, very little information is currently available about the role of *C. avidum* biofilms in PJI. Therefore, we investigated antibiotic susceptibility and virulence properties such as biofilm formation and fitness of *C. avidum* isolates recovered from deep tissues in PJI patients and compared them to isolates from healthy skin. We performed whole-genome sequence analysis of the core genome to examine the phylogenetic relationship between the cohort.

## MATERIALS AND METHODS

### Strain isolation and identification

The study (research project involving biological material and health-related personal data, clinical trial number: not applicable) includes 43 *C. avidum* isolates, either recovered from PJI ($n = 11$) from four European hospitals as part of a multicenter study supported by the European Study Group for Implant-Associated Infections of the European Society of Clinical Microbiology and Infectious Diseases or taken from healthy skin (HS) volunteers ($n = 32$) by scraping the skin with sterile blades. Skin scrapings were removed from blades and transferred into ESwab culture swabs (Copan, Brescia, Italy) (13). Swabs were cultured onto Schaedler-5% sheep blood agar plates (BioMérieux, Marcy l'Étoile, France) for 48 h at 37°C under anaerobic conditions and isolates were identified by MALDI-TOF MS (Vitek MS, BioMérieux, Marcy l'Étoile, France). Clinical data of *C. avidum* isolates recovered from PJI are shown in Table S1.

### Biofilm formation

Biofilm formation of all PJI and skin isolates was evaluated using a modified method of Stepanović et al. (14) as a static biofilm assay. Briefly, brain heart infusion (BHI) broth with 2% glucose was used with bacterial inocula of $10^7$ CFU/mL and incubated at 37°C under anaerobic conditions in 96-well plates. After 72 h of incubation, BHI broth was removed, and the wells were rinsed two times with methanol, and crystal violet was used for staining. The optical density (OD) of each well was measured at 570 nm using a microtiter plate reader. Each isolate was tested in triplicate. Isolates were divided into different categories based on OD values and a cut-off value (ODc) was established. The results were interpreted as: no biofilm producer (OD ≤ ODc), weak biofilm producer (ODc < OD ≤ 2 × ODc), moderate biofilm producer (2 × ODc < OD ≤ 4 × ODc), or strong biofilm producer (4 × ODc < OD).

## Susceptibility testing

Antibiotic susceptibility testing of 11 PJI-causing *C. avidum* isolates was performed. Antibiotics tested were amoxicillin-clavulanate, clindamycin, levofloxacin, linezolid, penicillin, rifampin, and vancomycin. The minimal inhibitory concentration (MIC) and the minimal bactericidal concentration (MBC) were determined with the broth microdilution method according to the protocols of the European Committee on Antimicrobial Susceptibility Testing (EUCAST) (15). Colonies of the isolates were suspended in Mueller Hinton cation-adjusted broth. Sterile round-bottomed 96-well plates were inoculated with 100 μL of Mueller-Hinton cation-adjusted broth containing the antimicrobial agent plus 100 μL of the bacterial suspension for obtaining a final inoculum of $10^4$ CFU per well and incubated under anaerobic conditions. After 48 h of incubation, the MIC was determined (the first antibiotic concentration where there was no turbidity). Subsequently, 50 μL of each well was transferred into a new plate containing 150 μL of Mueller Hinton cation-adjusted broth by using a modified flash microbiocide method (16). Plates were incubated for 2 days at 37°C under anaerobic conditions, and the MBC was determined (the first antibiotic concentration where there was no bacterial growth).

The minimal biofilm inhibitory concentration (MBIC) and the minimal biofilm eradication concentration (MBEC) were assessed following the protocols previously described by Coenye et al. (17). Briefly, colonies of the isolates were transferred into sterile phosphate-buffered saline (PBS), and the supernatant was adjusted to a turbidity of 0.5 ± 0.02 McFarland. A volume of 200 μL of the bacterial suspension was transferred into each well of a sterile, flat-bottomed, polystyrene 96-well plate tissue cultured. Plates were incubated in anaerobiosis jars (Oxoid Ltd., Thermo Fisher Scientific, Boston, MA, USA) to allow bacterial adhesion to the well bottom. Following 4 h of adhesion, the supernatant, containing the planktonic bacteria, was carefully removed. Each well was washed with 200 μL of sterile PBS to remove non-adherent cells. Then, 200 μL of clostridial nutrient medium (Sigma-Aldrich, Missouri, USA) was added, and the plates were incubated for 24 h at 37°C under anaerobic conditions to allow biofilm maturation. After incubation, the supernatant was again removed, and the plates were washed with 200 μL of sterile PBS. Each well was then filled with 100 μL of clostridial nutrient medium plus 100 μL of clostridial nutrient medium containing serial dilutions of the antimicrobial agent, and plates were further incubated. After 48 h of incubation in anaerobic conditions, the MBIC was determined (the first antibiotic concentration where there was no turbidity). After the MBIC assessment, the wells were washed with 200 μL of sterile PBS and each well was filled with 200 μL of fresh clostridial nutrient medium. The biofilm was mechanically disrupted by vigorously scraping the well bottom using sterile pipette tips, followed by homogenization. The MBEC was determined after 48 h of incubation (the first antibiotic concentration, where there was no bacterial growth).

The EUCAST resistance breakpoints for *C. acnes* were used to interpret antimicrobial susceptibility results (18). Levofloxacin and rifampin breakpoints have not been determined for anaerobic gram-positive bacteria.

Statistical analysis was performed using GraphPad Prism 8.4.3 (GraphPad Software, San Diego, CA, USA). Data were evaluated using a Wilcoxon nonparametric test to compare two groups. Statistical significance was set at $P$ values ≤ 0.05. Figure 3 was generated using R (19).

## Bacterial quantitative fitness analysis (BaQFA)

To assess reproductive fitness variations among *C. avidum* isolates, the BaQFA method was used (20). This approach involved spotting 96 bacterial cultures on an agar plate and capturing their growth over time through time-lapse photography with a BaQFA robot using an Arduino platform (21).

Eleven invasive isolates recovered from PJI and 31 superficial skin isolates were tested against one superficial skin strain as reference strain to analyze the difference in fitness. *C. avidum* strain PAVI-2017310081 was used as reference strain for all BaQFA experiments. *C. avidum* isolates were streaked out on Columbia sheep blood agar plates

and incubated at 37°C under anaerobic conditions for 3 days. Fresh bacterial colonies were scraped from the agar plates and diluted in PBS to an $OD_{600nm}$ of 0.10 (±0.01). The bacterial solution was diluted 1:10 in PBS, and 3 µL spots was spotted onto a rectangular single-well BHI agar plate with 20 mL agar medium in a grid pattern (each tested strain was grown in direct neighborhood of the competing reference strain). The agar plate was transferred into the BaQFA setup and incubated at 37°C under anaerobic conditions. Automated image capturing (every 30 min) was performed.

These images were analyzed using the BaColonyzer software to extract each colony's growth over time for detailed growth curves (21). Fitness was then derived from the parameters of a Gompertz growth model fitted to these curves as previously described (20). Taking into account the variability inherent in experimental runs, each of the runs of 96 culture spots per plate of the same strain comparisons was considered as a separate study with a potentially random difference in fitness estimate. Therefore, meta-analysis with a random effects model was used for the computation and plotting of relative fitness estimates and their confidence intervals (22), allowing for a comprehensive assessment of strain-specific fitness differences with the reference strain (PAVI-2017310081) in pair-wise comparisons.

## Genomic analyses

### DNA isolation and genome sequencing

For genomic DNA extraction, the Master Pure Gram-Positive DNA Purification Kit (Lucigen) was used as per the manufacturer's instructions. Concentration and purity of the isolated DNA were first checked with a NanoDrop ND-1000 (Peqlab, Erlangen, Germany); concentrations were determined using the Qubit dsDNA HS Assay Kit as recommended by the manufacturer (Life Technologies GmbH, Darmstadt, Germany). Illumina shotgun libraries were prepared using the Nextera XT DNA Sample Preparation Kit and subsequently sequenced on a MiSeq system using the v3 reagent kit with 600 cycles (Illumina, San Diego, CA, USA) as recommended by the manufacturer. Quality filtering was done with version 0.36 of Trimmomatic (23). Assembly was performed with version 3.13.0 of the SPAdes genome assembler software (24). Version 2.2.1 of Qualimap was used to validate the assembly and determine the sequence coverage (25). All genome sequences were deposited in GenBank, and the accession numbers are listed in Table S2.

### Bioinformatics tools and analyses

For phylogenomic analyses, the core genome was identified and aligned with the Parsnp program from the Harvest software package (26). Reliable core-genome SNPs identified by Parsnp were used for reconstruction of whole-genome phylogeny. Phylogenetic trees were visualized using the Interactive Tree Of Life (iTOL; https://itol.embl.de/). ResFinder (27) was used to identify (acquired) genes mediating antimicrobial resistance.

Proteinortho was used to identify intra- and interclade-specific gene content, applying a bidirectional blast approach (28). Orthologous proteins were identified with the following blast settings: coverage > 50% and identity > 50%. The following genomes were used to identify Clade 1- versus Clade 2-specific gene content differences: CI828_clade 1; CI878_clade1; CI882_clade1; PAVI-2017310081_clade1; PAVI-2017310082_clade1; PAVI-2017310120_clade1; HS4_clade2; HS7_clade2; HS9_clade2; PAVI-2017310084_clade2; PAVI-2017310145_clade2; and PAVI-2017310195_clade2. To identify potential differences between skin isolates and PJI isolates within Clade 1 the following genomes were compared: CI828_clade 1_PJI; CI878_clade1_PJI; CI882_clade1_ PJI; PAVI-2017310081_clade1_healthy skin; PAVI-2017310082_clade1_healthy skin; and PAVI-2017310120_clade1_healthy skin. Annotations were done with BV-BRC (29) and the KEGG tool BlastKOALA (30).

## RESULTS

### Genomic analysis

Eleven PJI isolates and 32 HS isolates were sequenced. The genome size of the isolates ranged from 2,471 to 2,728 kb, thus differing by a maximum of 257 kb. Annotation predicted coding sequences (CDS), ranging from 2,362 to 2,654, thus differing by a maximum of 292 CDS. Next, a core genome comparison was carried out. Two main phylogenetic clades (Clade 1 and Clade 2) within the *C. avidum* population could be detected (Fig. 1). Interestingly, all PJI isolates belonged to the same clade (Clade 1). This clade also contained isolates from other PJI cases, including isolates T13, T14, and T15 (8, 31) and FMS2275 and FMS4815 (32), as well as isolates associated with other diseases such as strain TM16, isolated from radical prostatectomy specimens (33). In contrast, HS isolates could be found in both clades, Clade 1 and Clade 2. We noticed that the genomes of Clade 2 isolates ($n = 6$) were on average 129 kb larger than the genomes of Clade 1 isolates ($n = 37$). Within Clade 1, genomes of skin isolates ($n = 26$) were on average 42 kb larger than genomes of PJI isolates ($n = 11$). We searched for consistent gene content differences between skin isolates and PJI isolates within Clade 1 (Table S3A); only very few CDS or CDS fragments were found to differ (Table S3B). Next, we searched for consistent gene content differences between Clade 1 and Clade 2 isolates (Table S4A). Here, 209 and 272 CDS (or CDS fragments) were found to be Clade 1- or Clade 2-specific, respectively (Table S4B and C).

### Biofilm

We compared biofilm formation of 11 isolates recovered from PJI with 32 isolates from HS (Fig. 2A). No statistically significant differences were found between the two groups. Furthermore, in the sub-analysis performed for Clade 1, the differences between the HS group and the PJI group were also not significant (Fig. 2B). According to the methodology used using a static biofilm assay, all isolates were biofilm producers. In both groups, most of the isolates were strong biofilm producers (93.7% skin group; 72.7% PJI group), and the rest of the isolates were moderate biofilm producers. No statistically significant differences in the proportions of strong and moderate biofilm producers in the two different groups were found ($P$-value 0.27).

### Antibiotic susceptibility

*C. avidum* isolates from PJIs displayed generally low MIC values for most antibiotics tested (Fig. 3). MIC and MBIC distributions were nearly identical for the majority of agents, with MBICs being slightly higher in some isolates, indicating limited biofilm tolerance. Rifampin showed the most potent activity against both planktonic and biofilm-embedded cells, with MIC, MBC, and MBIC values ≤0.125 mg/L in all isolates. Notably, MBEC values for rifampin remained low, with most isolates eradicated at ≤0.5 mg/L, contrasting sharply with other agents. For penicillin and levofloxacin, MBC and MBEC values were markedly higher than MIC/MBIC, indicating a limited bactericidal effect and poor biofilm eradication. Amoxicillin-clavulanic acid, linezolid, vancomycin, and clindamycin exhibited favorable MIC and MBC profiles; however, MBEC values were >32 mg/L in most cases, suggesting reduced efficacy in eradicating mature biofilms. One isolate (HOL 4) demonstrated clindamycin elevated MIC, MBC, MBIC, and MBEC values, consistent with the presence of the *erm(X)* resistance gene. Overall, rifampin was the only antibiotic to consistently exhibit low MIC, MBC, MBIC, and MBEC values, highlighting its superior activity against *C. avidum* in both planktonic and biofilm states.

### BaQFA

Since reproductive fitness of bacteria plays a major role in the evolution of antimicrobial resistance as well as persistence, which has implications in chronic infections, we

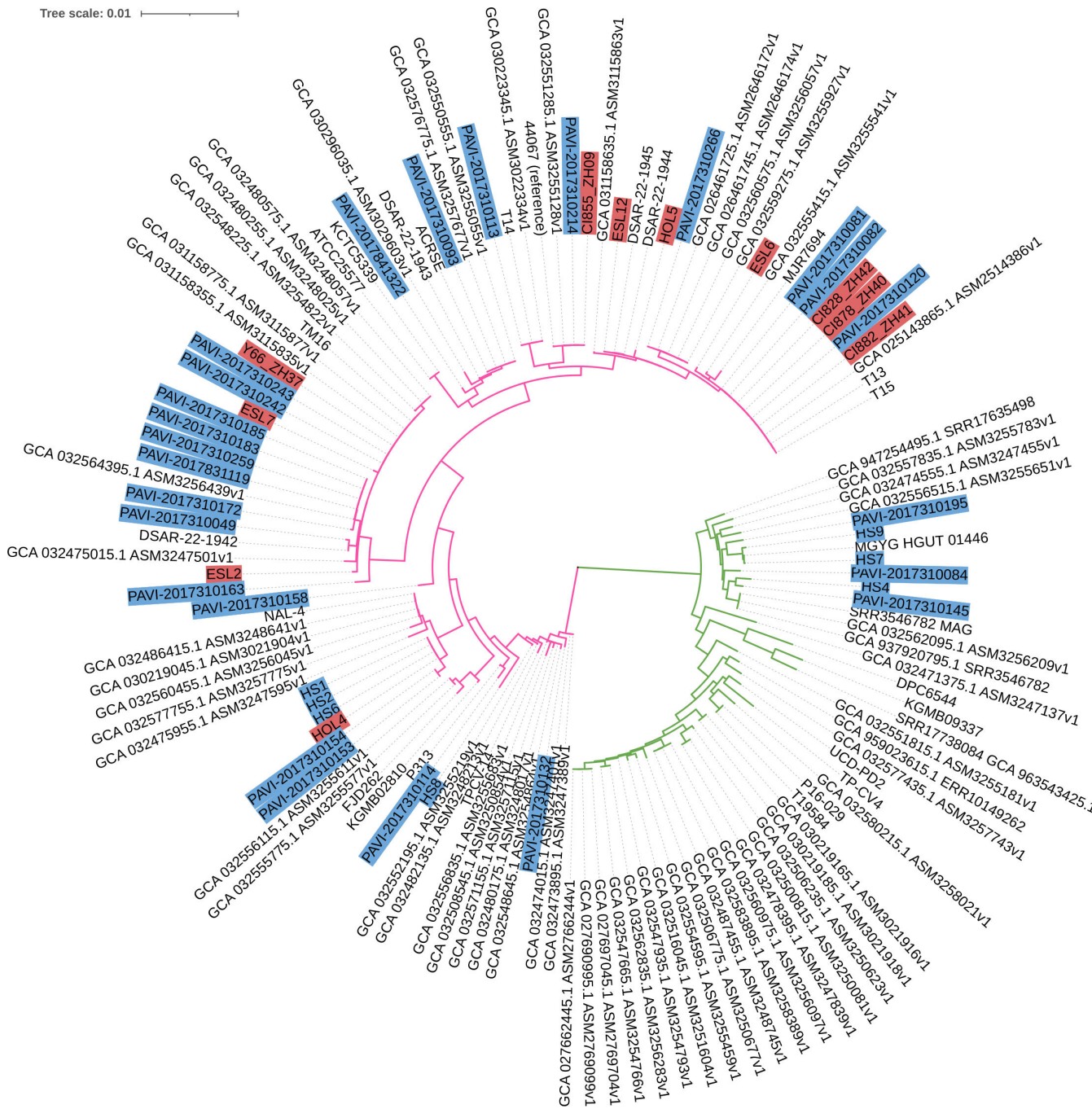

**FIG 1** Phylogenetic comparison of PJI and healthy skin isolates of *C. avidum* based on the core genome. Core genome-based single-nucleotide variant (SNV) analysis and phylogenetic reconstruction were done with Parsnp. The isolates studied here are highlighted in color; PJI isolates in red; isolates from healthy skin in blue. The other genomes were taken from GenBank (NCBI) (status February 2024). Two large clades can be distinguished. Clade 1 (pink) harbors all PJI isolates. Healthy skin isolates are distributed among Clade 1 (pink) and Clade 2 (green).

quantitatively assessed fitness differences between bacterial isolates. To do so, we used the BaQFA method (20), which allows us to define the relative competitive fitness (RCF) between two given isolates. We chose to assess the RCF of all isolates compared to one randomly selected reference isolate from HS. When including all strains (Clade 1 and Clade 2 strains), we detected significantly higher RCF for HS isolates as compared to isolates which caused PJI (*P* value 0.039) (Fig. 4A). However, when restricting the analysis to Clade 1 isolates only (the clade that contains PJI and HS isolates), this difference no

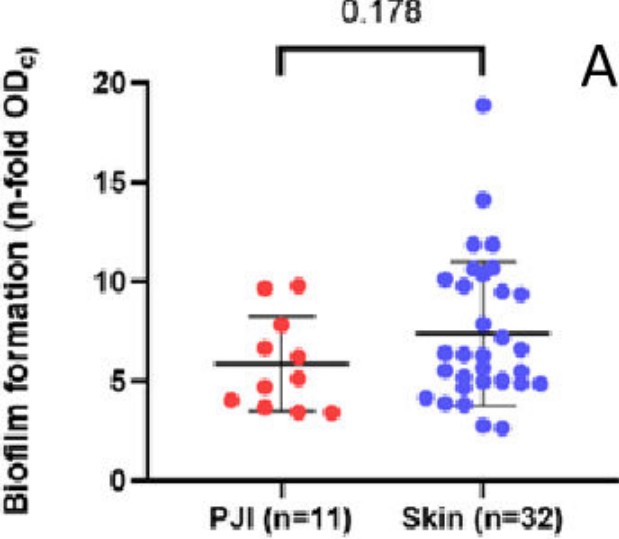

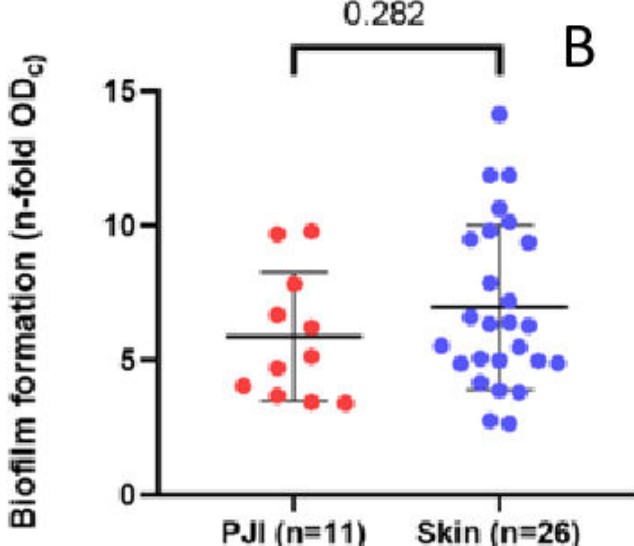

**FIG 2** Biofilm formation of healthy skin colonizers and PJI isolates of *C. avidum* using a modified method of Stepanović et al. (14) as a static biofilm assay. (A) Biofilm formation of all isolates (healthy skin colonizers, $n = 32$; PJI isolates, $n = 11$) was compared. (B) Biofilm formation of Clade 1 isolates was compared (healthy skin colonizers, $n = 26$; PJI isolates, $n = 11$). Statistical significance was assessed using the Mann-Whitney test following evaluation of data normality.

longer reached statistical significance, although a similar trend was observed ($P = 0.054$) (Fig. 4B). RCF values within the skin-derived group were notably heterogeneous. Upon exclusion of outlier isolates within the Clade 1 subset, the difference in RCF between PJI and skin isolates further diminished and was not statistically significant ($P = 0.099$). To explore potential clade-specific differences in bacterial fitness, we compared RCF between *C. avidum* isolates belonging to Clade 1 and Clade 2. No significant difference in RCF was observed between Clade 1 and Clade 2 isolates (median RCF: Clade 1 = X, Clade 2 = Y; $P = 0.8903$) (Fig. 4C).

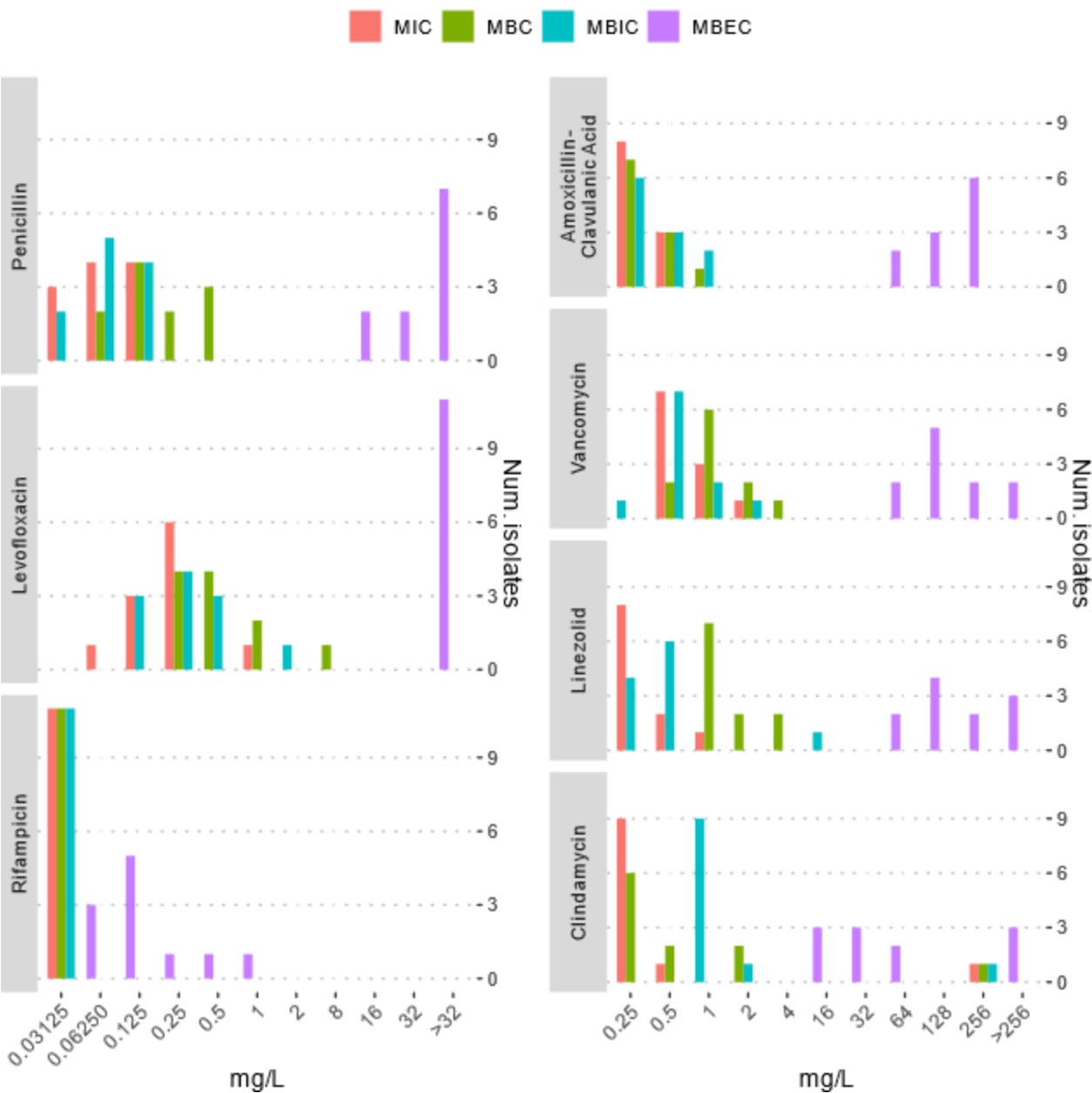

**FIG 3** Antibiotic susceptibility profiles of *Cutibacterium avidum* isolates (*n* = 11) from prosthetic joint infections. Minimal inhibitory concentrations (MICs), minimal bactericidal concentrations (MBCs), minimal biofilm inhibitory concentrations (MBICs), and minimal biofilm eradication concentrations (MBECs) were determined for penicillin, amoxicillin-clavulanic acid, clindamycin, linezolid, vancomycin, levofloxacin, and rifampin. Bars represent the number of isolates exhibiting a given concentration (mg/L) for each antibiotic and susceptibility parameter.

## DISCUSSION

To our knowledge, this is the most comprehensive comparative study of *C. avidum* isolates derived from PJIs and from HS, integrating analyses of phylogeny, biofilm formation, antibiotic susceptibility, and bacterial fitness.

Phylogenomic analysis revealed two distinct phylogenetic clades, Clade 1 and Clade 2, which may represent subspecies of *C. avidum*. Notably, all PJI isolates belonged exclusively to Clade 1, while skin isolates were distributed across both clades. This suggests that only Clade 1 isolates possess the potential to cause implant-associated infections.

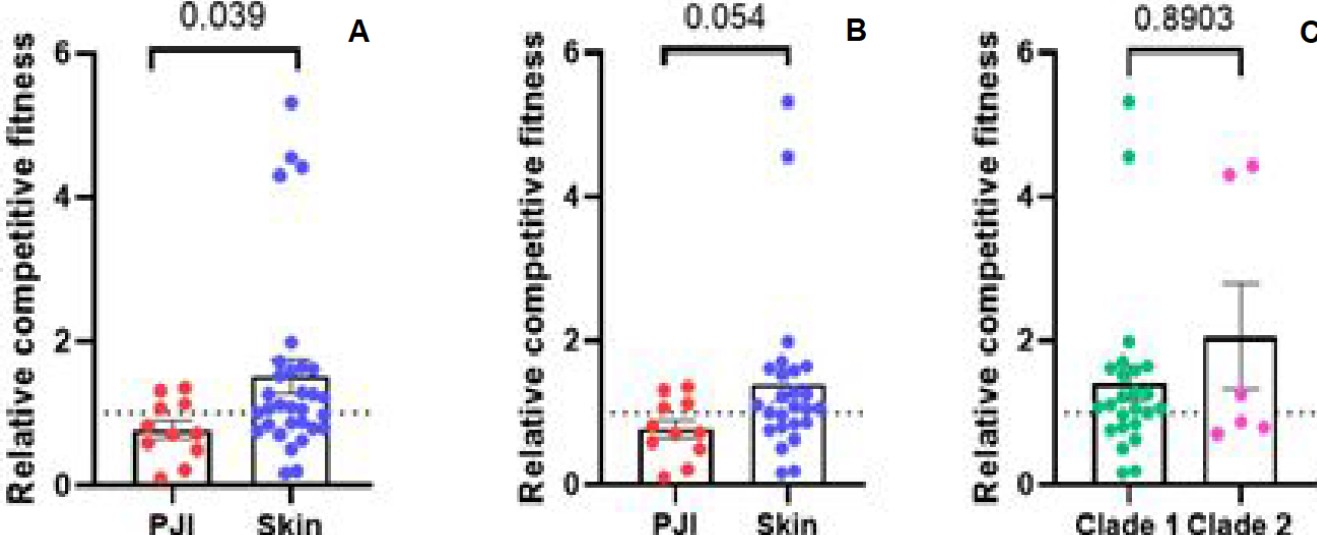

**FIG 4** Relative competitive fitness (RCF) of *C. avidum* isolates assessed using the BaQFA method. (A) RCF values were compared between isolates from all clades obtained from healthy skin (*n* = 32) and PJIs (*n* = 11). (B) RCF values were compared between Clade 1 isolates only. No statistical significance, but a trend was seen (*P* = 0.054). (C) RCF values were compared between Clade 1 and Clade 2 isolates. No significant difference was observed. The dashed horizontal line indicates the reference isolate set at RCF = 1. Plotted values are means from two independent biological repeats. Statistical significance was determined by Mann-Whitney test after testing for normal distribution.

Gene content comparison between Clade 1 and Clade 2 isolates revealed 209 and 272 clade-specific genes, respectively. The majority of genes could not be functionally annotated. Clade 1 specific functions were primarily associated with signaling and transport processes (Fig. S1). In contrast, Clade 2 isolates harbored genes enabling carbohydrate metabolism (e.g., sorbitol, galactitol, and myo-inositol) and *de novo* fatty acid biosynthesis via the type II fatty acid synthesis (FASII) pathway, which is absent in Clade 1. These metabolic differences may reflect niche-specific adaptation, suggesting that Clade 2 strains are better equipped to survive in nutrient-limited environments, whereas Clade 1 strains may be more dependent on host-derived nutrients such as fatty acids.

All *C. avidum* isolates in our study exhibited strong biofilm-forming capacity, irrespective of clinical origin or clade affiliation. This stands in contrast to previous findings in *C. acnes*, where most isolates formed only moderate or weak biofilms (34). The strong biofilm phenotype observed in *C. avidum* may be linked to a species-specific exopolysaccharide-like structure, which is absent in other *Cutibacterium* species (33) and may facilitate both surface adherence and antibiotic tolerance (31, 33).

MIC values observed in our study were consistent with previous data (8, 31, 35) and comparable to those reported for *C. acnes* in orthopedic implant-associated infections (36). One isolate (HOL 4) demonstrated high clindamycin MICs, associated with the *erm(X)* gene, a known resistance determinant in *Cutibacterium* spp (37). Importantly, no species-specific EUCAST or CLSI clinical breakpoints exist for *C. avidum*, and we therefore present our MIC data descriptively without categorizing isolates as "susceptible" or "resistant." Rifampin showed the lowest MIC, MBC, MBIC, and MBEC values, suggesting high *in vitro* activity against both planktonic and biofilm-associated bacteria. Nevertheless, the clinical efficacy of rifampin remains debated (38), and its use should be carefully balanced against the risk of adverse effects (39) and the lack of clear evidence from well-controlled studies.

Using BaQFA, we found that PJI isolates showed reduced relative competitive fitness compared to commensal skin isolates, when all isolates are taken into consideration (Clade 1 and Clade 2). A sub-analysis of fitness restricted to Clade 1 showed that

the difference between PJI and skin isolates was no longer statistically significant (*P* = 0.053), although the trend persisted. Given the known role of bacterial fitness in persistence and antimicrobial tolerance, this may represent an adaptive phenotype favoring chronic infection. Reduced fitness is consistent with a more dormant, slow-growing state, potentially resembling bacterial persisters (40). Such states are known to reduce susceptibility to antibiotics and immune clearance (41), thus promoting chronic infection. Our findings support the notion that biofilm-related persistence may be driven not only by extracellular matrix protection, but also by shifts in bacterial growth dynamics and metabolic activity (42–44). Whether reduced RCF is a cause or consequence of biofilm formation in *C. avidum* remains unclear. One hypothesis is that an adaptation favoring invasiveness may involve a trade-off in proliferative capacity. Alternatively, reduced growth could itself promote biofilm formation and persistence under the hostile conditions of the prosthetic joint microenvironment.

A major limitation of this study is the absence of species-specific clinical breakpoints for *C. avidum*. Our interpretations rely on descriptive MIC values and cannot inform treatment decisions directly. Additionally, the broth microdilution method used for susceptibility testing, though standard, may underestimate MICs for certain agents such as amoxicillin-clavulanic acid, as EUCAST has noted for *C. acnes*. Furthermore, while our isolate set represents the largest cohort studied to date, the sample size remains relatively small, limiting statistical power and generalizability.

In summary, *C. avidum* isolates from PJIs and HS show strong *in vitro* biofilm formation regardless of clinical origin. All PJI isolates belonged to Clade 1, suggesting this clade harbors the potential for pathogenicity. While antibiotic resistance was rare, biofilm-related persistence and reduced bacterial fitness may contribute to treatment challenges. Our findings underscore the need for species-specific clinical breakpoints, better functional understanding of phylogenetic clades, and further investigation into biofilm-driven phenotypes in *C. avidum*-associated infections.

## ACKNOWLEDGMENTS

The authors thank the ESGIAI group for the study of Cutibacterium infections: Rihard Trebse (Orthopaedic Hospital Valdoltra, Ankaran, Slovenia), Mitja Rak (National Laboratory of Health, Environment and Food, Koper, Slovenia), Marjan Wouthuyzen-Bakker (University Medical Center Groningen, University of Groningen, Groningen, Netherlands). We thank Jared Liu for submitting the genome sequences of the Zurich *C. avidum* isolates to GenBank. We thank Francisco J. Pardo-Palacios for designing the figures.

This work was funded by the Swiss Life Jubiläumsstiftung (Y.A.), the grant CIBERIN-FEC-CIBER de Enfermedades Infecciosas CB21/13/00043 (J.E.). and the "Fabrikant Vilhelm Pedersen og Hustrus Legat" (by the recommendation from the Novo Nordisk Foundation) (no. 30658) (H.B.).

L.S.V.: Methodology, validation, formal analysis, investigation, data curation, writing—original draft, visualization. J.P.: Investigation, data curation, writing—original draft, visualization. P.M.F.: Methodology, software, formal analysis, data curation, writing—review and editing. T.A.S.: Validation, data curation, writing—review and editing. L.B.: Investigation, writing—review and editing. S.D.B.: Conceptualization, methodology, supervision, funding acquisition, writing—review and editing, data curation. H.B.: Methodology, resources, investigations, supervision, funding acquisition, writing—review and editing. J.E.: Conceptualization, methodology, resources, supervision, funding acquisition, writing—review and editing. Y.A.: Conceptualization, resources, methodology, supervision, funding acquisition, writing—review and editing, corresponding author.

## AUTHOR AFFILIATIONS

[1]Department of Clinical Microbiology, IIS—Fundación Jiménez Díaz, UAM, Madrid, Spain

²CIBERINFEC-CIBER de Enfermedades Infecciosas, Instituto de Salud Carlos III, Madrid, Spain

³Department of Dermatology, University Hospital Zurich, Zurich, Switzerland

⁴Department of Infectious Diseases, University Hospital Zurich, University of Zurich, Zurich, Switzerland

⁵Department of General Internal Medicine, Bern University Hospital (Inselspital), University of Bern, Bern, Switzerland

⁶Department of Biomedicine, Aarhus University, Aarhus, Denmark

⁷Hospital Zollikerberg, Zollikerberg, Switzerland

## AUTHOR ORCIDs

Holger Brüggemann  http://orcid.org/0000-0001-7433-0190

Jaime Esteban  http://orcid.org/0000-0002-8971-3167

Yvonne Achermann  http://orcid.org/0000-0001-7747-937X

## FUNDING

| Funder | Grant(s) | Author(s) |
|---|---|---|
| Jubiläumsstiftung von Swiss Life | | Yvonne Achermann |
| Centro de Investigación Biotecnológica en Red de Enfermedades Infecciosas (CIBERINFEC) | CB21/13/00043 | Jaime Esteban |
| Fabrikant Vilhelm Pedersen og Hustrus Legat (VPL) | | Holger Brüggemann |

## AUTHOR CONTRIBUTIONS

Llanos Salar Vidal, Investigation, Writing – review and editing | Julia Prinz, Investigation, Writing – review and editing | Pascal M. Frey, Investigation, Writing – review and editing | Tiziano A. Schweizer, Investigation, Writing – review and editing | Laura Böni, Investigation, Writing – review and editing | Silvio D. Brugger, Investigation, Writing – review and editing | Holger Brüggemann, Conceptualization, Data curation, Formal analysis, Funding acquisition, Investigation, Methodology, Supervision, Validation, Visualization, Writing – original draft, Writing – review and editing | Jaime Esteban, Conceptualization, Resources, Writing – review and editing | Yvonne Achermann, Conceptualization, Data curation, Formal analysis, Funding acquisition, Methodology, Project administration, Supervision, Writing – original draft, Writing – review and editing

## DATA AVAILABILITY

All data generated or analyzed during this study are included in this published article. The data sets are available in the GenBank repository and are accessible here: https://www.ncbi.nlm.nih.gov/bioproject/PRJNA1078905/ and https://www.ncbi.nlm.nih.gov/bioproject/PRJNA729908/ and https://www.ncbi.nlm.nih.gov/bioproject/PRJNA380511/.

## ETHICS APPROVAL

The cantonal ethics committee of Zurich, Switzerland, approved the study protocol (BASEC-Nr 2016-01017, BASEC Nr. 2019-00924) for the use of non-genetic data and bacterial samples, with a waiver to include consent of patients, and the approval of local ethical committees of Spain, Netherlands, and Slovenia.

## ADDITIONAL FILES

The following material is available online.

## Supplemental Material

**Supplemental material (Spectrum00515-25-s0001.docx).** Tables S1 and S2; Fig. S1.
**Table S3 (Spectrum00515-25-s0002.xlsx).** Proteinortho comparison of genomes within clade 1 (3 PJI isolates and 3 skin isolates).
**Table S4 (Spectrum00515-25-s0003.xlsx).** Clade 2-specific CDS.

## Open Peer Review

**PEER REVIEW HISTORY (review-history.pdf).** An accounting of the reviewer comments and feedback.

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
