## [Reviewer comments · Microbiology Spectrum]

Microbiology Spectrum

Bacterial skin colonization with a specific *Cutibacterium avidum* clade as a risk factor for periprosthetic joint infections - a multi-center study

Llanos Salar-Vidal, Julia Prinz, Pascal Frey, Tiziano Schweizer, Laura Boeni, Silvio Brugger, Holger Brüggemann, Jaime Esteban, and Yvonne Achermann

Corresponding Author(s): Yvonne Achermann, University Hospital of Zurich

Review Timeline:

Submission Date:	February 19, 2025
Editorial Decision:	May 27, 2025
Revision Received:	July 8, 2025
Editorial Decision:	July 18, 2025
Revision Received:	August 19, 2025
Accepted:	August 22, 2025

Editor: Gillian Tarr

Reviewer(s): The reviewers have opted to remain anonymous.

Transaction Report:

DOI: <https://doi.org/10.1128/spectrum.00515-25>

Re: Spectrum00515-25 (**Bacterial skin colonization with a specific *Cutibacterium avidum* lineage as a risk factor for periprosthetic joint infections - a multi-center study**)

Dear Dr. Llanos Salar-Vidal:

Thank you for the privilege of reviewing your work. Below you will find my comments, instructions from the Spectrum editorial office, and the reviewer comments.

Please respond to both the comments copied below and those in the attached review.

Revision Guidelines

Sincerely,
Gillian Tarr
Editor
Microbiology Spectrum

Reviewer #1 (Comments for the Author):

Vidal et al. have submitted a publication titled "Bacterial Skin Colonization with a Specific *Cutibacterium avidum* Lineage as a Risk Factor for Periprosthetic Joint Infections - A Multi-Center Study." This study investigates the biofilm formation, antibiotic susceptibility, and metabolic fitness of 11 *C. avidum* isolates from periprosthetic joint infection (PJI) patients and 32 isolates from healthy skin, collected from four hospitals across Europe. Whole-genome sequencing was performed on all isolates. The results

indicate that PJI-associated strains belong to a distinct clade, exhibit reduced metabolic fitness, and form robust biofilms. Despite their antibiotic susceptibility, these characteristics may contribute to therapeutic failure. This study represents the most extensive analysis of *C. avidum* isolates to date, spanning multiple hospitals, and could serve as a valuable reference for future research. Additionally, it may prompt a reassessment of therapeutic strategies. The study is methodologically sound and well-documented. However, some minor revisions could improve clarity:

Lines 140-141: The MBEC methodology is not clearly described; additional details are needed.

Table 1: Define the terms MIC50, MIC90, MBIC50, and MBIC90 for clarity.

Reviewer #2 (Comments for the Author):

This manuscript addresses an important clinical question regarding the characterization of *C. avidum* isolates from prosthetic joint infections. The methodological approach is sound and the multi-parameter characterization provides valuable data. However, the phylogenetic structure revealing two distinct clades has significant implications for the interpretation of all comparative analyses, and this issue must be adequately addressed.

The main recommendations for revision include: (1) restricting comparative analyses to phylogenetically appropriate groups or providing stratified analyses by clade, (2) correcting antimicrobial susceptibility data presentation according to current EUCAST guidelines, (3) re-evaluating fitness analysis results in light of the phylogenetic structure, and (4) moderating overstated claims that are not supported by the presented data.

Vidal et al. have submitted a publication titled "*Bacterial Skin Colonization with a Specific Cutibacterium avidum Lineage as a Risk Factor for Periprosthetic Joint Infections – A Multi-Center Study.*" This study investigates the biofilm formation, antibiotic susceptibility, and metabolic fitness of 11 *C. avidum* isolates from periprosthetic joint infection (PJI) patients and 32 isolates from healthy skin, collected from four hospitals across Europe. Whole-genome sequencing was performed on all isolates. The results indicate that PJI-associated strains belong to a distinct clade, exhibit reduced metabolic fitness, and form robust biofilms. Despite their antibiotic susceptibility, these characteristics may contribute to therapeutic failure. This study represents the most extensive analysis of *C. avidum* isolates to date, spanning multiple hospitals, and could serve as a valuable reference for future research. Additionally, it may prompt a reassessment of therapeutic strategies. The study is methodologically sound and well-documented. However, some minor revisions could improve clarity:

- **Lines 140–141:** The MBEC methodology is not clearly described; additional details are needed.
- **Table 1:** Define the terms MIC50, MIC90, MBIC50, and MBIC90 for clarity.

Review

Bacterial skin colonization with a specific *Cutibacterium avidum* lineage as a risk 2 factor for periprosthetic joint infections - a multi-center study

Overall Assessment

This study investigates *Cutibacterium avidum* isolates from prosthetic joint infections and skin isolates. I recommend accept with major revision.

The manuscript employs a sound methodological approach for isolate characterization. The authors have included a biofilm formation assay, which is relevant for prosthetic joint infection studies, and the antimicrobial susceptibility testing encompasses multiple parameters (MIC, MBC, MBIC, and MBEC), providing insights into both planktonic and biofilm-associated resistance. The inclusion of bacterial fitness assays represents a valuable addition to the characterization panel.

Major Concerns

Phylogenetic Structure and Comparative Analysis The phylogenetic analysis reveals two distinct clades that appear to represent potentially different subspecies of *C. avidum*. Clade 1 contains both prosthetic joint infection (PJI) isolates and skin isolates, while Clade 2 contains exclusively skin isolates (n=6). This phylogenetic separation raises important concerns about the validity of the comparative analyses presented.

Given that Clade 2 isolates may represent a distinct subspecies, all comparative analyses between PJI isolates and skin isolates should exclude the six skin isolates belonging to Clade 2. The current approach of comparing all skin isolates as a single group against PJI isolates may be confounding subspecies-level differences with niche-specific adaptations. This could lead to misinterpretation of which phenotypic differences are truly associated with prosthetic joint infection versus those that simply reflect taxonomic diversity within the *C. avidum* complex.

The authors should either: (1) restrict comparative analyses to isolates within Clade 1 only, or (2) present stratified analyses that clearly distinguish between the two clades and discuss the implications of this phylogenetic structure for their conclusions about pathogenic potential and clinical relevance.

Antimicrobial Susceptibility Testing and Data Presentation The authors reference EUCAST breakpoint table version 12, while the current version is 15. Critically, the current EUCAST guidelines (v15) do not provide clinical breakpoints for *C. avidum*, with breakpoints available only for *C. acnes*. This absence of species-specific breakpoints significantly impacts the interpretation and presentation of antimicrobial susceptibility data.

Several methodological and presentation issues need to be addressed:

1. **Inappropriate methodology:** For amoxicillin-clavulanic acid, EUCAST specifically advises against broth microdilution (BMD) when testing *C. acnes* as MIC values may be falsely low. The authors should acknowledge this limitation or use alternative methods.
2. **Misleading statistical presentation:** Reporting MIC50 and MIC90 values for a collection of only 11 isolates is not meaningful and may be misleading. Instead, the authors should present a clear table showing the distribution of MIC values for each antibiotic, indicating how many isolates exhibited each specific MIC value.

3. **Inappropriate terminology:** Without established clinical breakpoints for *C. avidum*, the authors should avoid using terms such as "susceptible" and "resistant." They should follow EUCAST recommendations for reporting data when clinical breakpoints are unavailable, focusing on MIC distributions and using appropriate cautionary language.
4. **Missing context:** The clinical relevance of the MIC values should be discussed with appropriate caveats about the lack of species-specific breakpoints and the limitations this places on clinical interpretation.

Bacterial Quantitative Fitness Analysis The bacterial fitness data presented in Figure 2 shows four apparent outliers among the skin isolates with notably different competitive fitness values. It is crucial to determine which phylogenetic clade these outlier isolates belong to. If these outliers correspond to the Clade 2 skin isolates (which may represent a different subspecies as discussed above), their inclusion in the comparative analysis could be driving apparent differences between PJI and skin isolates.

The authors should clarify:

1. Which phylogenetic clade do the four fitness outliers belong to?
2. When these outliers are excluded from the analysis, is there still a statistically significant difference in competitive fitness between PJI isolates and skin isolates?
3. If the outliers belong to Clade 2, this would further support the argument for excluding Clade 2 isolates from all comparative analyses.

The fitness analysis should be reanalyzed with appropriate stratification by phylogenetic clade to determine whether observed fitness differences are truly associated with clinical niche adaptation or simply reflect subspecies-level variation. This is essential for drawing valid conclusions about the pathogenic potential and clinical relevance of different *C. avidum* populations.

Minor Concerns

Terminology Throughout the manuscript, the authors should use "isolates" rather than "strains" when referring to the *C. avidum* samples included in this study. The term "strain" typically refers to well-characterized, genetically defined variants that are maintained in culture collections, while "isolate" is more appropriate for clinical specimens that have been isolated and characterized but may not represent distinct genetic lineages. This terminology correction should be applied consistently throughout the text, figures, and tables. The authors should also maintain consistent terminology when discussing phylogenetic groupings. It is recommended to use "clades" consistently throughout the manuscript rather than alternating between "lineage" and "clade."

Overstated Claims In the "Importance" section, the statement "this study highlights the growing significance of *C. avidum* in periprosthetic joint infections (PJI)" (line 51-52) is not supported by the data presented. The authors provide no epidemiological data or trends demonstrating that the incidence of *C. avidum* PJI is actually increasing. The statement should be revised to reflect what the study actually demonstrates - the characterization of *C. avidum* isolates from PJI cases - rather than making unsupported claims about temporal trends in clinical significance. Similarly, the authors' claim that "a specific *C. avidum* lineage associated with PJI was identified" (line 53-54) is not supported by the phylogenetic analysis. The data actually show that most PJI isolates are closely related to skin isolates within the same clade. What is the SNP distance between the most divergent PJI isolate and its closest

related skin isolate? This information would help evaluate whether the genetic differences are sufficient to support claims of PJI-specific adaptation.

Methods and Figure Details The Methods section lacks information about which publicly available genomes were included in the phylogenetic analysis and the criteria used for their selection. This information is essential for reproducibility and interpretation of the phylogenetic relationships.

The figure legend for Figure 3 should be updated to specify the total number of genomes included in the analysis (both study isolates and publicly available genomes).

Specific Comments

Line 42: The statement "PJI isolates had in average smaller genomes than healthy skin isolates" should be removed or revised. Does this observation hold true when the six isolates from Clade 2 are excluded from the analysis? Given the phylogenetic structure identified, this comparison may be confounded by subspecies-level differences rather than reflecting true niche-specific adaptations.

Line 200 - Figure 1: There appear to be more than 32 dots in the left figure, which does not correspond to the total number of skin isolates. Please verify the figure accuracy and ensure it matches the actual sample size.

Line 208: Setting aside the broader issue of using resistance terminology without appropriate breakpoints, there appears to be an inconsistency regarding clindamycin resistance. The authors state that only one isolate was resistant to clindamycin, using *S. aureus* breakpoints (R >0.25 mg/L). However, the data suggest there are isolates with MIC values of both 0.5 mg/L and >256 mg/L, which would both be considered resistant by these criteria. Additionally, how was the *erm(X)* gene detected? Genotypic resistance determination is not described in the Methods section, yet resistance genes are mentioned in the results.

Lines 221-224 and Figure 2: According to Figure 2, no isolate appears to exhibit lower comparative fitness than the reference isolate (y-axis starts at 0). This seems unexpected for a competitive fitness assay where one would anticipate some variation both above and below the randomly selected reference. Please clarify this observation and verify the data presentation.

Lines 229-230: See comment for line 42 regarding genome size comparisons.

Concluding Comments

This manuscript addresses an important clinical question regarding the characterization of *C. avidum* isolates from prosthetic joint infections. The methodological approach is sound and the multi-parameter characterization provides valuable data. However, the phylogenetic structure revealing two distinct clades has significant implications for the interpretation of all comparative analyses, and this issue must be adequately addressed.

The main recommendations for revision include: (1) restricting comparative analyses to phylogenetically appropriate groups or providing stratified analyses by clade, (2) correcting antimicrobial susceptibility data presentation according to current EUCAST guidelines, (3) re-evaluating fitness analysis results in light of the phylogenetic structure, and (4) moderating overstated claims that are not supported by the presented data.

With these revisions, this study has the potential to make a meaningful contribution to our understanding of *C. avidum* diversity and clinical relevance. The authors are encouraged to

address these concerns, as the underlying data appear valuable for the clinical microbiology community.

Point by Point

Bacterial skin colonization with a specific *Cutibacterium avidum* lineage as a risk factor for periprosthetic joint infections - a multi-center study

Llanos Salar-Vidal (corr-auth) , Dr. Julia Prinz , Pascal Frey , Tiziano A. Schweizer , Laura Boeni , Prof. Silvio D. Brugger , Prof. Holger Brüggemann , Dr. Jaime Esteban , Dr. Yvonne Achermann

Dear Editors,

We would like to thank you and the reviewers for the constructive and helpful comments.

All points raised have been addressed and integrated into the revised manuscript. In response to the new analyses performed, both the abstract and the discussion have been substantially restructured and rewritten to reflect the updated findings and interpretations.

We hope that the revised version meets your expectations and look forward to your feedback.

Sincerely,

Yvonne Achermann

Reviewer 1:

Vidal et al. have submitted a publication titled "Bacterial Skin Colonization with a Specific *Cutibacterium avidum* Lineage as a Risk Factor for Periprosthetic Joint Infections - A Multi-Center Study." This study investigates the biofilm formation, antibiotic susceptibility, and metabolic fitness of 11 *C. avidum* isolates from periprosthetic joint infection (PJI) patients and 32 isolates from healthy skin, collected from four hospitals across Europe. Whole-genome sequencing was performed on all isolates. The results indicate that PJI-associated strains belong to a distinct clade, exhibit reduced metabolic fitness, and form robust biofilms. Despite their antibiotic susceptibility, these characteristics may contribute to therapeutic failure. This study represents the most extensive analysis of *C. avidum* isolates to date, spanning multiple hospitals, and could serve as a valuable reference for future research. Additionally, it may prompt a reassessment of therapeutic strategies. The study is methodologically sound and well-documented. However, some minor revisions could improve clarity:

Lines 140-141: The MBEC methodology is not clearly described; additional details are needed.

Table 1: Define the terms MIC50, MIC90, MBIC50, and MBIC90 for clarity.

Response. We thank the reviewer for this valuable comment. The methodology section has been thoroughly revised and rewritten to improve clarity and ensure that each step of the protocol is described in a more structured and comprehensible manner.

Table 1 was replaced by a new figure according to reviewer 1 and reviewer 2 comments.

Reviewer 2:

Review

Bacterial skin colonization with a specific *Cutibacterium avidum* lineage as a risk 2 factor for periprosthetic joint infections - a multi-center study

Overall Assessment

This study investigates *Cutibacterium avidum* isolates from prosthetic joint infections and skin isolates. I recommend accept with major revision.

The manuscript employs a sound methodological approach for isolate characterization. The authors have included a biofilm formation assay, which is relevant for prosthetic joint infection studies, and the antimicrobial susceptibility testing encompasses multiple parameters (MIC, MBC, MBIC, and MBEC), providing insights into both planktonic and biofilm-associated resistance. The inclusion of bacterial fitness assays represents a valuable addition to the characterization panel.

Response: We thank reviewer 2 for the possibility to review our manuscript

Major Concerns

Phylogenetic Structure and Comparative Analysis The phylogenetic analysis reveals two distinct clades that appear to represent potentially different subspecies of *C. avidum*. Clade 1 contains both prosthetic joint infection (PJI) isolates and skin isolates, while Clade 2 contains exclusively skin isolates (n=6). This phylogenetic separation raises important concerns about the validity of the comparative analyses presented.

Given that Clade 2 isolates may represent a distinct subspecies, all comparative analyses between PJI isolates and skin isolates should exclude the six skin isolates belonging to Clade 2. The current approach of comparing all skin isolates as a single group against PJI isolates may be confounding subspecies-level differences with niche-specific adaptations. This could lead to misinterpretation of which phenotypic differences are truly associated with prosthetic joint infection versus those that simply reflect taxonomic diversity within the *C. avidum* complex.

The authors should either: (1) restrict comparative analyses to isolates within Clade 1 only, or (2) present stratified analyses that clearly distinguish between the two clades and discuss the implications of this phylogenetic structure for their conclusions about pathogenic potential and clinical relevance.

Response. Thank you for your thoughtful and constructive comment regarding the phylogenetic structure of the *Cutibacterium avidum* isolates and its implications for our comparative analyses.

We agree that the observed separation into two distinct clades — with Clade 1 comprising both PJI and skin isolates and Clade 2 containing only skin isolates — raises important considerations regarding potential subspecies/clade-level differences within the *C. avidum* complex.

In response to your suggestion, we have revised our comparative analyses to address this issue. Specifically, we now:

- 1. Restrict our primary comparative analyses to isolates within Clade 1 only, which includes both skin and PJI isolates. This ensures that differences identified between PJI and skin isolates are not confounded by underlying phylogenetic divergence.**

We also compared the genomes of skin and PJI isolates of clade 1 only, but found no differences in gene content. With other words: there is no specific gene gain or loss in PJI isolates compared to skin isolates (all within clade 1). This could mean that all isolates of clade 1 have the potential to cause a PJI. This has been added to the results and discussions.

2. Additionally, we now present stratified analyses that compare isolates of clade 1 (PJI/skin isolates) to isolates of Clade 2 (skin isolates). These results are discussed in the revised Results and Discussion sections, where we explicitly address the potential implications of phylogenetic structure for pathogenic potential and clinical relevance.

Regarding genomic comparisons we identified clade 1- and clade 2-specific genes, respectively. Functional interpretations are not straight-forward, given the lack of knowledge regarding virulence (e.g. PJI causing traits) and metabolism of this species. However, we added a few predicted differences based on genome content analysis in the discussion. It is likely that the different gene content of clade1 and clade 2 isolates reflects niche-specific adaptation.

Antimicrobial Susceptibility Testing and Data Presentation The authors reference EUCAST breakpoint table version 12, while the current version is 15. Critically, the current EUCAST guidelines (v15) do not provide clinical breakpoints for *C. avidum*, with breakpoints available only for *C. acnes*. This absence of species-specific breakpoints significantly impacts the interpretation and presentation of antimicrobial susceptibility data.

Several methodological and presentation issues need to be addressed:

1. **Inappropriate methodology:** For amoxicillin-clavulanic acid, EUCAST specifically advises against broth microdilution (BMD) when testing *C. acnes* as MIC values may be falsely low. The authors should acknowledge this limitation or use alternative methods.

Response: Thank you for this important observation. We acknowledge that the current EUCAST version (v15) does not provide clinical breakpoints for *Cutibacterium avidum*, and that breakpoints for *C. acnes* were referenced as a surrogate in our study. We agree that this introduces uncertainty in the interpretation of antimicrobial susceptibility results.

Furthermore, we recognize that EUCAST explicitly advises against the use of broth microdilution (BMD) for testing amoxicillin-clavulanic acid in *C. acnes*, due to the potential for falsely low MICs. Although BMD remains a widely used method, we appreciate the importance of highlighting its limitations for this specific drug-organism combination.

In response, we have added a dedicated paragraph in the Discussion section (line 337) addressing these limitations. We now explicitly state the absence of *C. avidum*-specific breakpoints, the reliance on surrogate breakpoints from *C. acnes*, and the limitations of using BMD for amoxicillin-clavulanic acid. We also suggest that future studies explore alternative methods such as gradient diffusion testing.

Misleading statistical presentation: Reporting MIC₅₀ and MIC₉₀ values for a collection of only 11 isolates is not meaningful and may be misleading. Instead, the authors should present a clear table showing the distribution of MIC values for each antibiotic, indicating how many isolates exhibited each specific MIC value.

Response. We appreciate the reviewer's suggestion. In response, we have removed the MIC₅₀ and MIC₉₀ values and created a new figure that presents the full distribution of MIC, MBIC, and MBEC values for each antibiotic. The figure (Figure 2) now clearly shows the number of isolates corresponding to each specific value.

Inappropriate terminology: Without established clinical breakpoints for *C. avidum*, the authors should avoid using terms such as "susceptible" and "resistant." They should follow EUCAST recommendations for reporting data when clinical breakpoints are unavailable, focusing on MIC distributions and using appropriate cautionary language.

Response: Thank you for this important comment. We have revised the entire section on antibiotic susceptibility to avoid inappropriate terminology such as "susceptible" or "resistant," in accordance with EUCAST recommendations. The revised text now focuses on MIC distributions and uses cautious, descriptive language. In addition, we have included a new figure (Figure 2) illustrating the antibiotic susceptibility profiles based on MIC, MBC, MBIC, and MBEC values.

Missing context: The clinical relevance of the MIC values should be discussed with appropriate caveats about the lack of species-specific breakpoints and the limitations this places on clinical interpretation.

Response: Thank you for this valuable comment. We fully acknowledge that clinical breakpoints specific to *Cutibacterium avidum* are currently not established by EUCAST or CLSI. We have therefore added a paragraph to the Discussion section addressing the limited clinical interpretability of MIC values in this context. The revised text emphasizes that our findings are descriptive and should not be used to guide therapy directly, but rather to inform future research and potential breakpoint development

Bacterial Quantitative Fitness Analysis The bacterial fitness data presented in Figure 2 shows four apparent outliers among the skin isolates with notably different competitive fitness values. It is crucial to determine which phylogenetic clade these outlier isolates belong to. If these outliers correspond to the Clade 2 skin isolates (which may represent a different subspecies as discussed above), their inclusion in the comparative analysis could be driving apparent differences between PJI and skin isolates.

The authors should clarify:

1. Which phylogenetic clade do the four fitness outliers belong to?
2. When these outliers are excluded from the analysis, is there still a statistically significant difference in competitive fitness between PJI isolates and skin isolates? no
3. If the outliers belong to Clade 2, this would further support the argument for excluding Clade 2 isolates from all comparative analyses.

The fitness analysis should be reanalyzed with appropriate stratification by phylogenetic clade to determine whether observed fitness differences are truly associated with clinical niche adaptation or simply reflect subspecies-level variation. This is essential for drawing valid conclusions about the pathogenic potential and clinical relevance of different *C. avidum* populations.

Response:

Thank you for this important and detailed observation. In our main analysis, we included all isolates from Clade 1 and Clade 2. In response to this comment, we have now conducted a subanalysis restricted to Clade 1 isolates only.

1. **The four apparent fitness outliers among the skin isolates belong to both Clade 1 and Clade 2.**
2. **When these outliers were excluded from the Clade 1 subset, the difference in relative competitive fitness between PJI and skin isolates was no longer statistically significant.**
3. **These findings support the notion that the inclusion of Clade 2 isolates—potentially representing a distinct subspecies—may confound comparative analyses. We have therefore clarified in the manuscript that Clade 1-only analyses are more appropriate**

for assessing phenotypic differences between clinical (PJI) and commensal (skin) isolates. The corresponding text and figure legend have been updated accordingly.

Minor Concerns

Terminology Throughout the manuscript, the authors should use "isolates" rather than "strains" when referring to the *C. avidum* samples included in this study. The term "strain" typically refers to well-characterized, genetically defined variants that are maintained in culture collections, while "isolate" is more appropriate for clinical specimens that have been isolated and characterized but may not represent distinct genetic lineages. This terminology correction should be applied consistently throughout the text, figures, and tables. The authors should also maintain consistent terminology when discussing phylogenetic groupings. It is recommended to use "clades" consistently throughout the manuscript rather than alternating between "lineage" and "clade."

Response: Thank you for pointing out the important distinction regarding terminology. We agree that the term "isolates" is more appropriate than "strains" when referring to clinical *C. avidum* samples in the context of this study. We have carefully reviewed the entire manuscript and revised the text, tables, and figure legends accordingly to consistently use "isolates."

Additionally, we have standardized the terminology for phylogenetic groupings. We now use the term "clades" consistently throughout the manuscript, replacing instances where we previously used "lineage" or other alternative terms. We appreciate your guidance in improving the clarity and precision of our manuscript.

Overstated Claims In the "Importance" section, the statement "this study highlights the growing significance of *C. avidum* periprosthetic joint infections (PJI)" (line 51-52) is not supported by the data presented. The authors provide no epidemiological data or trends demonstrating that the incidence of *C. avidum* PJI is actually increasing. The statement should be revised to reflect what the study actually demonstrates - the characterization of *C. avidum* isolates from PJI cases - rather than making unsupported claims about temporal trends in clinical significance. Similarly, the authors' claim that "a specific *C. avidum* lineage associated with PJI was identified" (line 53-54) is not supported by the phylogenetic analysis. The data actually show that most PJI isolates are closely related to skin isolates within the same clade. What is the SNP distance between the most divergent PJI isolate and its closest

Response: Thank you for this important and well-taken comment. We agree that the statement in the "*Importance*" section suggesting a "growing significance" of *C. avidum* in PJI is not directly supported by our dataset, as we did not assess epidemiological trends or incidence over time. We have therefore revised this statement to more accurately reflect the scope of our study, emphasizing the genomic and phenotypic characterization of *C. avidum* isolates from PJI cases rather than implying an increase in prevalence.

Specific Comments

Line 42: The statement "PJI isolates had in average smaller genomes than healthy skin isolates" should be removed or revised. Does this observation hold true when the six isolates from Clade 2 are excluded from the analysis? Given the phylogenetic structure identified, this comparison may be confounded by subspecies-level differences rather than reflecting true niche-specific adaptations.

Response. To overcome this problem raised by the review, two separate comparisons were done:

PJI isolates (n=11) versus skin isolates (n=26) of clade 1: a small difference in genome size was observed (genomes of skin isolates in average 42 kb larger).

Clade 1 isolates (n=37) versus clade 2 isolates (n=6): genomes of clade 2 isolates were in average 129 kb larger. The abstract and the results were modified accordingly.

Line 200 - Figure 1: There appear to be more than 32 dots in the left figure, which does not correspond to the total number of skin isolates. Please verify the figure accuracy and ensure it matches the actual sample size.

Response. We thank the reviewer for pointing this out. The figure has been revised, and it now accurately reflects the total number of skin isolates (n = 32).

Line 208: Setting aside the broader issue of using resistance terminology without appropriate breakpoints, there appears to be an inconsistency regarding clindamycin resistance. The authors state that only one isolate was resistant to clindamycin, using *S. aureus* breakpoints (R >0.25 mg/L). However, the data suggest there are isolates with MIC values of both 0.5 mg/L and >256 mg/L, which would both be considered resistant by these criteria. Additionally, how was the *erm(X)* gene detected? Genotypic resistance determination is not described in the Methods section, yet resistance genes are mentioned in the results.

Response. Thank you for bringing this to our attention. In response, we have revised the way of susceptibility data presentation. We have removed the table 1 (that included MIC₅₀, MIC₉₀, etc.), which we agree may be misleading given the limited number of isolates. A new figure showing the distribution of MIC, MBC, MBIC and MBEC values for each antibiotic, indicating the number of isolates corresponding to each concentration, providing a more accurate representation of the data. We now specify that the susceptibility interpretations were made using the EUCAST version 15.0 breakpoints for *Cutibacterium acnes*, acknowledging the limitations of applying these breakpoints to a different species. Additionally, in material and methods section (Bioinformatic tools and analysis), it is now specified that we have used ResFinder to determine *erm(X)* gene detection.

Lines 221-224 and Figure 2: According to Figure 2, no isolate appears to exhibit lower comparative fitness than the reference isolate (y-axis starts at 0). This seems unexpected for a competitive fitness assay where one would anticipate some variation both above and below the randomly selected reference. Please clarify this observation and verify the data presentation.

Response: Thank you for this helpful observation. We would like to clarify that the reference isolate was set to a relative competitive fitness (RCF) of 1. All other isolates were assessed relative to this reference. Accordingly, isolates with an RCF below 1 have a lower fitness than the reference isolate. The y-axis in Figure 2 starts at 0 (not at 1), which may have caused confusion. To improve clarity, we have revised the figure to include a dashed horizontal line at RCF = 1, marking the fitness of the reference isolate. The figure legend and main text have also been updated accordingly.

Lines 229-230: See comment for line 42 regarding genome size comparisons.

Concluding Comments

This manuscript addresses an important clinical question regarding the characterization of *C. avidum* isolates from prosthetic joint infections. The methodological approach is sound and the multi-parameter characterization provides valuable data. However, the phylogenetic structure revealing two distinct clades has significant implications for the interpretation of all comparative analyses, and this issue must be adequately addressed.

The main recommendations for revision include: (1) restricting comparative analyses to phylogenetically appropriate groups or providing stratified analyses by clade, (2) correcting antimicrobial susceptibility data presentation according to current EUCAST guidelines, (3) re-evaluating fitness analysis results in light of the phylogenetic structure, and (4) moderating overstated claims that are not supported by the presented data.

With these revisions, this study has the potential to make a meaningful contribution to our understanding of *C. avidum* diversity and clinical relevance. The authors are encouraged to address these concerns, as the underlying data appear valuable for the clinical microbiology community.

Re: Spectrum00515-25R1 (**Bacterial skin colonization with a specific *Cutibacterium avidum* lineage as a risk factor for periprosthetic joint infections - a multi-center study**)

Dear Dr. Yvonne Achermann:

Thank you for the privilege of reviewing your work. Below you will find my comments and instructions from the Spectrum editorial office.

Thank you for responding to the comments provided by the reviewers. Some additional modifications are needed to fully address the concerns of Reviewer 2. Your secondary analysis suggests that the reviewer's concern that the comparative analyses are confounded by clade is valid. There are also some miscellaneous changes still needed. Please make the following modifications to proceed with publication:

1. All comparative analyses should be presented for Clade 1 alone, and these should be given primacy. This is particularly true for the fitness analysis. Currently, the combined analysis and the Clade 1-only analysis are presented in separate sections in both the Results and Discussion. These should be presented together in both places so that readers are not misled about what's going on in the system by the combined analysis. The results from the combined analysis are also called out in the first paragraph of the Discussion. Given that the association is confounded by clade, it should not be presented as a primary result.
2. Page 12, line 240, you use the term clindamycin-resistant isolate. As you changed all other uses of resistant/susceptible, it appears this one was missed.
3. The Figure 2 legend says, "Antibiotic susceptibility profiles of *Cutibacterium avidum* isolates (n = X) from prosthetic joint infections." You need to replace the "X" with a number.

Revision Guidelines

Sincerely,
Gillian Tarr
Editor
Microbiology Spectrum

Editor in Chief
Microbiology Spectrum

Yvonne Achermann, PD Dr. med.

FMH Innere Medizin und
Infektiologie

Wissenschaftliche Mitarbeiterin
Universitätsspital Zürich
Rämistrasse 100
CH-8091 Zürich
044 396 74 18
yvonne.achermann@usz.ch

Zürich, 11.8.2025/YA

www.usz.ch

Resubmission *Manuscript*: « *Bacterial skin colonization with a specific *Cutibacterium avidum* clade as a risk factor for periprosthetic joint infections - a multi-center study* »

We thank you and the reviewers for the constructive feedback on our manuscript. Below we summarise how we have addressed Reviewer 1's comments:

- 1. Presentation of comparative analyses within Clade 1**
We have rearranged the text and figures so that the genomic analysis is presented first, followed by all downstream analyses (including fitness testing) restricted to Clade 1. This ensures that the Clade 1 analyses are given primacy and are presented alongside any combined analyses for clarity. We have also removed emphasis on the combined analysis from the first paragraph of the Discussion and clarified its secondary role, thereby avoiding potential misinterpretation.
- 2. Terminology consistency for antimicrobial susceptibility**
On page 12, line 240, we have replaced "clindamycin-resistant isolate" with the updated terminology consistent with the rest of the manuscript.
- 3. Correction to Figure 2 legend**
We have replaced the placeholder "X" in the Figure 2 legend with the correct number of isolates.

We believe these changes address the reviewer's concerns and improve the clarity and focus of our manuscript. We are grateful for the helpful suggestions and hope the revised version will now be acceptable for publication.

Sincerely,

Yvonne Achermann,
on behalf of all co-authors

PD Dr. med. Yvonne Achermann

Re: Spectrum00515-25R2 (**Bacterial skin colonization with a specific *Cutibacterium avidum* lineage as a risk factor for periprosthetic joint infections - a multi-center study**)

Dear Dr. Yvonne Achermann:

Thank you for your diligence in responding to the comments.

Your manuscript has been accepted, and I am forwarding it to the ASM production staff for publication. Your paper will first be checked to make sure all elements meet the technical requirements. ASM staff will contact you if anything needs to be revised before copyediting and production can begin. Otherwise, you will be notified when your proofs are ready to be viewed.

Sincerely,
Gillian Tarr
Editor
Microbiology Spectrum